

# Transkingdom network reveals bacterial players associated with cervical cancer gene expression program

Khiem Chi Lam[1,2,*], Dariia Vyshenska[1,*], Jialu Hu[1,3], Richard Rosario Rodrigues[1], Anja Nilsen[4], Ryszard A. Zielke[1], Nicholas Samuel Brown[1], Eva-Katrine Aarnes[4], Aleksandra E. Sikora[1], Natalia Shulzhenko[5], Heidi Lyng[4] and Andrey Morgun[1]

[1] College of Pharmacy, Oregon State University, Corvallis, OR, USA
[2] Center for Cancer Research, National Cancer Institute, Bethesda, MD, USA
[3] School of Computer Science, Northwestern Polytechnical University, Xi'an, China
[4] Institute for Cancer Research, Norwegian Radium Hospital, Oslo University Hospital, Oslo, Norway
[5] Carlson College of Veterinary Medicine, Oregon State University, Corvallis, OR, USA
* These authors contributed equally to this work.

Corresponding authors
Heidi Lyng,
Heidi.Lyng@rr-research.no
Andrey Morgun,
andriy.morgun@oregonstate.edu

## ABSTRACT

Cervical cancer is the fourth most common cancer in women worldwide with human papillomavirus (HPV) being the main cause the disease. Chromosomal amplifications have been identified as a source of upregulation for cervical cancer driver genes but cannot fully explain increased expression of immune genes in invasive carcinoma. Insight into additional factors that may tip the balance from immune tolerance of HPV to the elimination of the virus may lead to better diagnosis markers. We investigated whether microbiota affect molecular pathways in cervical carcinogenesis by performing microbiome analysis via sequencing 16S rRNA in tumor biopsies from 121 patients. While we detected a large number of intra-tumor taxa (289 operational taxonomic units (OTUs)), we focused on the 38 most abundantly represented microbes. To search for microbes and host genes potentially involved in the interaction, we reconstructed a transkingdom network by integrating a previously discovered cervical cancer gene expression network with our bacterial co-abundance network and employed bipartite betweenness centrality. The top ranked microbes were represented by the families *Bacillaceae*, *Halobacteriaceae*, and *Prevotellaceae*. While we could not define the first two families to the species level, *Prevotellaceae* was assigned to *Prevotella bivia*. By co-culturing a cervical cancer cell line with *P. bivia*, we confirmed that three out of the ten top predicted genes in the transkingdom network (lysosomal associated membrane protein 3 (LAMP3), STAT1, TAP1), all regulators of immunological pathways, were upregulated by this microorganism. Therefore, we propose that intra-tumor microbiota may contribute to cervical carcinogenesis through the induction of immune response drivers, including the well-known cancer gene LAMP3.

## INTRODUCTION

Cervical cancer remains the fourth most common cancer in women worldwide, and the world's second highest cause of female cancer mortality (*Ginsburg et al., 2017*). Persistent infection with high-risk human papillomavirus (hrHPV) is a causative factor in cervical carcinogenesis (*Walboomers et al., 1999*). The HPV vaccination is expected to decrease the cancer incidence, however, in Central and Eastern Europe and developing countries where there is a lack of systematic screening and vaccination programs, morbidity and mortality are anticipated to remain high. Thus, cervical carcinogenesis will remain a major threat to women's health. Most women who are infected with HPV never develop cancer. Hence, persistent HPV infection is necessary but may not be sufficient to trigger cancer development. A better understanding of the additional factors required for carcinogenesis would improve identification of high-risk cases and represents a major step toward personalized medicine in cervical cancer (*Shulzhenko et al., 2014*).

According to the current model of cervical carcinogenesis, hrHPV exists in the episomal state within *basal cells* in the epithelium during early disease onset (*Munoz et al., 2006*). The viral gene E2, while expressed, suppresses expression of E6 and E7 viral oncogenes. The longer the infection persists, the more HPV integrates into the host genome. Upon integration, the open reading frame of E2 is disrupted which reduces the control of E6 and E7 and promotes cell proliferation (*Shulzhenko et al., 2014*). Therefore, the key event that turns chronic HPV infection into cancer is elimination of the episomal form of virus as it is the only source of the oncosuppressor E2. The host immune system may eliminate episomal HPV, giving the cells with an integrated form of virus a growth advantage (*Herdman et al., 2006*). Although insufficient antiviral immunity enables persistent HPV infection at an early state, proceeding to the integrated state seems to require activation of the woman's immune system. Chromosomal amplifications in the infected cells have recently been identified as one of the key sources of driver genes upregulation in this disease (*Mine et al., 2013*). However, these genomic alterations cannot fully explain increased expression of immune genes in invasive carcinoma. Additional factors that may tip the balance from making the immune system tolerate HPV to eliminate virus are so far unknown.

Microbiota can have a significant role in disease, contributing to development of metabolic (diabetes, enteropathy associated with common variable immunodeficiency, etc.) and immune (inflammatory bowel disease, asthma, allergy) disorders as well as cancer (colorectal, gastric, lung, pancreatic etc.) (*Ege et al., 2011*; *Farrell et al., 2012*; *Fujimura & Lynch, 2015*; *Geng et al., 2013*; *Greer et al., 2016*; *Hosgood et al., 2014*; *Knights, Lassen & Xavier, 2013*; *Marchesi et al., 2011*; *Shulzhenko et al., 2017*; *Stokholm et al., 2018*; *Uemura et al., 2001*). Microbial communities have also been found to specifically contribute to virus induced carcinogenesis (*Vyshenska et al., 2017*). In particular, multiple studies confirm vaginal dysbiosis (bacterial vaginosis) to be a risk factor for HPV infection (*Gillet et al., 2011*) and its progression (*Gillet et al., 2012*; *Oh et al., 2015*). Studies of vaginal (*Champer et al., 2017*) and cervical (*Vyshenska et al., 2017*) microbiota have found

associations between changes in microbial community (e.g., overall diversity, abundances of particular taxa) and cancer development. Furthermore, the location of this cancer suggests a possible role of common members of cervico-vaginal microbiota in cervical carcinogenesis. While almost 25 years ago, bacteria cultured from cervical cancer biopsies were proposed to contribute to cancer progression, we are still far from understanding the role of non-viral microbes in this disease (*Mikamo et al., 1993*).

In this study, we aimed to investigate whether microbiota may affect molecular pathways in cervical carcinogenesis. We reconstructed a transkingdom network that integrates microbiome and host transcriptome data in tumor samples from patients to infer key bacterial players. We demonstrate the significance of this approach by identifying several bacterial candidates and show that one of them (*Prevotella bivia)* upregulates a well-known human cancer driver, lysosomal associated membrane protein 3 (LAMP3).

## MATERIALS AND METHODS

### Patients

Tumor specimens were retrieved from 123 patients with locally advanced squamous cell carcinoma of the uterine cervix. One to four biopsies were taken at different locations of the tumor at the time of diagnosis, immediately frozen in liquid nitrogen, and stored at −80 °C. DNA and RNA from different biopsies of the same tumor were pooled. The clinical protocol was approved by the Regional Committee for Medical Research Ethics in southern Norway (REC no. S-01129). Written informed consent was obtained from all patients (Form S1). DNA was isolated according to a standard protocol with proteinase K, phenol, chloroform, and isoamylalcohol (*De Angelis et al., 1999*). Purified DNA quality and concentration were assessed using the Quant-iT™ PicoGreen® dsDNA Assay Kit (Life Technologies, Carlsbad, CA, USA).

### Gene expression

Gene expression profiling of 123 tumors used in this study was performed using the Illumina HumanWG-6 v3 Expression BeadArrays with approximately 48,000 transcripts (Illumina Inc., San Diego, CA, USA). All samples had more than 50% tumor cells in hematoxylin and eosin stained sections. This selection may have led to some bias in the results, but was chosen to reduce the influence of different normal cell proportion across the samples. Total RNA was isolated by the use of Trizol reagent (Life Technologies, Carlsbad, CA, USA) followed by LiCl precipitation. Hybridization, scanning, signal extraction, and normalization were performed as described (*Lando et al., 2009*).

### Bacterial DNA quantification

Bacterial content was quantified using QuantiFast SYBR Green mix (Qiagen, Germantown, MD, USA) and universal bacterial primers, UniF340 (5′-ACTCCTACGGG AGGCAGCAGT) and UniR514 (5′-ATTACCGCGGCTGCTGGC). Standards were created from serial dilutions of extracted DNA from bacteria grown from mouse cecum contents.

## 16S rRNA library preparation and sequencing

For MiSeq Illumina sequencing, total genomic DNA was subjected to PCR amplification targeting the 16S rRNA variable region 4 (v4) using the bacterial primers 515F/806R. Test reactions were performed with samples with varying amounts of bacterial DNA to determine the threshold for successful library preparation, defined as a positive band on an agarose gel. A total of 40–250 ng of total template DNA were used for each PCR reaction, performed in triplicates per sample. Each set of triplicate PCR reactions was pooled and purified using the MinElute PCR Purification Kit (Qiagen, Germantown, MD, USA). The pools were checked for proper band size of 382 bp by gel electrophoresis using 2% agarose pre-cast E-gels (Life Technologies, Carlsbad, CA, USA). Negative controls consisted of samples without template for DNA extraction and PCR amplification. The pools were then quantified using the Qubit dsDNA BR Assay Kit (Life Technologies, Carlsbad, CA, USA). Barcoded amplicons were pooled at equal volumes and concentrations (two µL of five ng/µL DNA). The total pool was sequenced using the Illumina MiSeq 2000 sequencing platform at the Center for Genome Research and Biocomputing at Oregon State University to generate pair-ended 250 nt reads. The dataset generated and analyzed during the current study is available in Sequence Read Archive under accession no. SRP131188.

## Processing of raw 16S rRNA reads

Raw forward-end FASTQ reads from the Illumina sequencing output were quality filtered, demultiplexed, and analyzed using quantitative insights into microbial ecology (QIIME) (*Caporaso et al., 2010*). Reads were quality filtered using default QIIME parameters; reads with a Phred quality score of <20, ambiguous base calls, and fewer than 187 nt (75% of 250 nt) of consecutive high-quality base calls were discarded. Additionally, truncation occurred on reads with three consecutive low-quality bases. The samples were demultiplexed using 12 bp barcodes, allowing for a maximum of 1.5 errors in the barcode.

Reads were clustered using UCLUST (*Edgar, 2010*) at 97% similarity into operational taxonomic units (OTUs) at QIIME default parameters. A representative set of sequences from each OTU were selected for taxonomic identification by selecting the cluster seeds (first read assigned to that OTU). Representative sequences for each OTU were aligned using BLAST (*e*-value < 0.001) to Greengenes (version 13.8) OTU reference sequences (97% similarity) to obtain taxonomy assignments. OTUs were filtered for singletons (only found in one sample) and relative abundance was quantified by dividing raw read counts by total number of reads for each sample. Alpha diversity rarefaction curves using the Shannon index and a heatmap of OTU frequencies were obtained from QIIME scripts using default parameters.

## Comparing bacterial communities from different body sites
### Human microbiome project data

High-quality FASTQ reads for region v3 and v5 were obtained (ftp://public-ftp.ihmpdcc.org/HMQCP/seqs_v35.fna.gz). Female patients from the

first visit were retained if they had samples from vagina (posterior_fornix) and at least stool or skin (left and right of antecubital fossa and retroauricular crease) to allow multi-site comparisons for the same patient. In case there were multiple samples from the same site of a patient, we retained the sample with highest number of sequences assigned to OTUs. This resulted a total of 380 samples from 77 females.

### Healthy cervix

Raw FASTQ data for 17 HPV negative samples was obtained from European Nucleotide Archive, Study PRJEB1872, and used as healthy cervix samples. High-quality reads (Phred score >19) were retained using QIIME's split_libraries_fastq.py command and a Phred offset of 33.

To allow comparisons between studies, reads for the 455 samples (Table S2) were assigned to OTUs at 97% sequence similarity using UCLUST and closed reference OTU picking. The reverse strand match was enabled for the human microbiome project (HMP) data during OTU picking. The studies were merged using common OTU IDs. Singleton OTUs were removed. The combined OTU table had 455 samples and 2,516,046 sequences assigned to 444 OTUs (Table S2). The table was rarefied with a sequence threshold of 1,000 sequences (Table S2) and used for diversity analysis. Beta diversity was calculated using weighted and unweighted unifrac and used for PCoA analysis. Alpha diversities were compared using Shannon diversity index.

Cervical cancer samples with greater than 1,000 sequences (based on the above threshold used for rarefaction) assigned to OTUs were retained and singleton OTUs were removed. The resulting OTU table had 432 OTUs for 52 cervical cancer samples (Table S2), was relativized and used to create a heat tree at the genus level using Metacoder (*Foster, Sharpton & Grunwald, 2017*).

## OTU collapsing

An initial quality check of de novo cervical cancer OTUs were made against healthy vaginal samples from the HMP (as described below). OTU representative sequences, OTU table, and mapping file for 16S rRNA v3–v5 sequencing were downloaded from the HMP QIIME community profiling datasets publicly available (*The Human Microbiome Project Consortium, 2012b*). Representative sequences from cervical cancer de novo picked OTUs were aligned to HMP representative sequences using USEARCH. Sequences with identity matches greater than 0.97 were collapsed together, keeping HMP OTU identification, taxonomy, and representative sequences as appropriate; if multiple cervical cancer OTUs matched, their read counts were summed for each sample. These new "collapsed" OTUs were used for subsequent analyses.

## Transkingdom network reconstruction

Pairwise Spearman correlations were calculated for each OTU with mean relative abundance >0.5%. Spearman correlations were additionally calculated between top OTUs and 738 differentially expressed genes (DEGs) (quantile-normalized gene expression microarray, raw data available in Gene Expression Omnibus under accession no. GSE68339) found in previous cancer gene regulatory network (*Mine et al., 2013*).

Correlations with $p < 0.001$ and FDR $< 0.1$ were imported into igraph R package for network reconstruction. Gene-OTU correlations were integrated with OTU–OTU correlations along with the previously published gene–gene network. Multiple edges and self-loops were removed. Visualization was performed in Cytoscape.

Bipartite betweenness centrality (BiBC) is the measure of probability for nodes belonging to one subnetwork to be bottlenecks in the transfer of signal to the nodes in another subnetwork, and vice versa. It was calculated as previously described (*Dong et al., 2015*) between the microbial subnetwork and each DEG subnetwork as well as all DEGs. Relativized BiBC values were calculated by dividing the value of a particular node to the sum of all nodes in that metric. Relativized BiBC values were then log transformed as followed:

$$log2\left(\left(BiBC * 10^6\right) + 1\right)$$

### *Prevotella* OTU alignment

Representative sequences were extracted for all original cancer de novo OTUs that were collapsed into OTU_97.1949. These sequences were aligned to SILVA 16S rRNA sequences for *Prevotella* using USEARCH. Matches with length >200 bp and mismatch ≤40 bp were kept for quantifications. For each species, the numbers of hits for each representative sequence were summed after normalization to the total number of hits for that representative sequence. This method was used due to multiple database matches for each given representative sequence due to high similarity and low specificity of the query sequence.

### Human cell culture

HeLa cervical cancer cell line was acquired from ATCC and maintained in modified Eagle's medium (EMEM) with Earle's balanced salt solution, L-glutamine, and non-essential amino acids, without Calcium. EMEM was supplemented with 5% heat-inactivated fetal bovine serum and 1% penicillin streptomycin solution (PEST). Frozen cells stored in 10% DMSO in liquid nitrogen were thawed in a 37 °C water bath until thawed (~3 min), added to four mL of fresh media, and centrifuged for 3 min at 2,500 rpm to pellet cells. The media solution was aspirated, and cells were resuspended in 15 mL of fresh media. Cells were grown in 5% $CO_2$ at 37 °C in 75 cm² treated, vented cap flat bottom culture flasks. Cells were passaged at approximately 80% confluency using three mL of 0.25% Trypsin supplemented with 2.21 mM EDTA without sodium bicarbonate to detach adherent cells.

### Bacteria cell culture

The following bacteria were used in this study: *P. bivia* (ATCC 29303) and *Lactobacillus crispatus* (ATCC 33197). All bacteria were grown on Brucella blood agar plates supplemented with hemin and vitamin K (Hardy Diagnostics, Santa Maria, CA, USA) at 37 °C in anaerobic conditions using the GasPak™ EZ Container System (BD Diagnostics, Sparks, MD, USA). Cultures were stored in 25% glycerol at −80 °C. Bacterial plates were restreaked once every 2–3 days as needed for experiments.

## Human-bacteria co-culture

Bacterial cultures were taken from frozen stocks, plated on Brucella blood agar plates, and restreaked after 3 days. After incubation for 2 days, bacteria were suspended in three mL of PBS and quantified by optical density at OD600 nm; correlations between OD600 and colony forming units were made prior to the experiments. HeLa cells were resuspended in EMEM media without PEST, counted using a hemocytometer, seeded at 75,000 cells/well in 24-well flat bottom culture plates, and incubated at 37 °C in 5% $CO_2$ 24 h prior to bacterial treatment. Bacteria were adjusted to appropriate concentration for 40 μL treatments at a multiplicity of infection of 10 in replicates of three or more, using sterile PBS as the negative control. Bacteria-treated HeLa cells were incubated at 37 °C in anaerobic conditions using GasPak™ EZ Anaerobe Container System Sachets for either 24 h, followed by on-plate lysis using RLT Lysis Buffer (Qiagen, Germantown, MD, USA) and lysate storage at −80 °C.

## RT-quantitative PCR for co-culture experiment

RNA was extracted from cell lysate using the RNeasy Mini Kit (Qiagen, Germantown, MD, USA) with on-column DNase digestion. RNA was quantified using Qubit RNA BR Assay Kit (Life Technologies, Carlsbad, CA, USA) and reverse transcribed using qScript cDNA Synthesis Kit (Quantabio, Beverly, MA, USA). RT-quantitative PCR (qPCR) was performed using PerfeCTA SYBR Green FastMix (Quantabio, Beverly, MA, USA) and human gene primers (Table S5). RT-qPCR set up was as follows: sample was heated to 95 °C, followed by 40 cycles of 95 °C for 10 s and 60 °C for 30 s. Fold change was calculated by normalizing to 18S rRNA housekeeping gene.

## RESULTS

Varying amounts of bacterial DNA were detected by qPCR in 121 out of 123 tumor samples from cervical cancer patients (Table S1). Sufficient amounts for 16S rRNA library preparation and sequencing was found in 58 samples. After 16S rRNA sequencing and quality filtering, we obtained 3,975,755 high-quality reads with a median of 87,320 reads/sample.

As a first step of analysis, we evaluated our cancer microbiome data in relation to publicly available datasets for adjacent healthy body sites (*Audirac-Chalifour et al., 2016*; *Di Paola et al., 2017*; *Gajer et al., 2012*; *Hong et al., 2016*; *Lee et al., 2013*; *Ling et al., 2010*; *Liu et al., 2013*; *Ravel et al., 2011*; *Si et al., 2017*) and sites sampled by the HMP (*The Human Microbiome Project Consortium, 2012a*, *2012b*). While such analysis is limited since the samples were not matched (i.e., not the same individuals), it provides general information about the microbiota in cervical cancer compared to other tissues. Unsupervised principal coordinate analysis using common OTUs showed that the cervical cancer samples had a different microbiome composition than samples from healthy cervix, vagina, stool, or skin samples (Fig. 1A). Compared to healthy uterine cervix and vagina, the cancer microbiome showed increased alpha diversity (Fig. S1) and high abundance of genera such as *Prevotella* (27.3%), *Fusobacterium* (17%), and *Peptoniphilus* (10.8%) as well as a low abundance of *Lactobacillus* (0.5%) (Figs. 1B and 1C and Table S2).

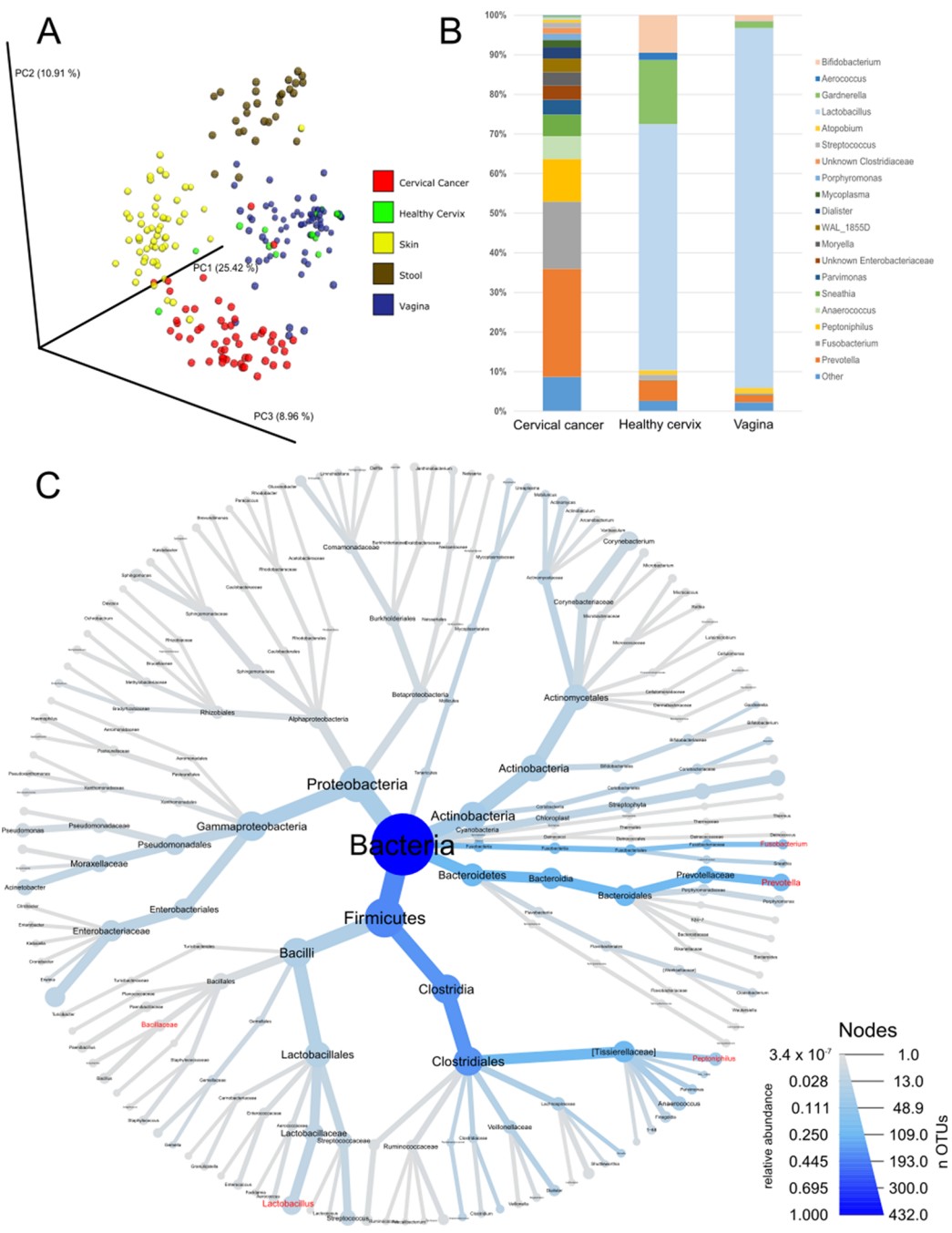

**Figure 1 Community composition in cervical cancer and healthy adjacent sites.** (A) PCoA of unweighted unifrac comparing microbiota of samples from cervical cancer ($n = 52$), healthy cervix ($n = 17$), vagina ($n = 76$), stool ($n = 28$), and skin ($n = 55$). (B) Bar chart of mean relative abundance of genera in cervical cancer biopsies, cytobrush from healthy ectocervical mucosa, and swab from healthy posterior vaginal fornix. Genera are arranged in descending order of mean relative abundance in cervical cancer samples from bottom to top of the bar chart. Genera with mean relative abundance <0.5% across the three sites are grouped into "Other" found at bottom of bar chart. (C) Phylogenetic tree indicating the relationship and mean relative abundance (blue color intensity) of various genera in cervical cancer samples. The size of node and its label indicate the number of OTUs belonging to that taxonomy.

To investigate whether the bacteria influence molecular pathways in the tumors, we built a transkingdom network that integrated microbial abundance and cancer gene expression networks. First, OTUs with an abundance of at least 0.5% were selected, and a bacterial co-abundance network that contained 38 nodes and 54 edges was reconstructed. For the human portion of the transkingdom network, we selected the network based on meta-analysis of transcriptomes from five cohorts of cervical cancer patients (*Shulzhenko et al., 2014*). In that study, we reconstructed a gene expression meta-network, identifying three pathways with upregulated cell cycle and proinflammatory/antiviral genes, and downregulated epithelial cell differentiation genes as key features of cervical carcinogenesis. To integrate the microbial and human subnetworks we calculated correlations between microbial abundances and gene expression from the cancer network measured in the same tumor samples as bacteria. The network contained 21 taxa and 698 human genes connected by 19 transkingdom edges (Fig. 2A).

The inflammatory/antiviral genes are the most plausible targets of regulation by bacteria. In addition, antiviral gene expression is of special interest in cervical cancer due to its involvement in elimination of episomal HPV that leads to tumor growth (*Pett et al., 2006*; *Vyshenska et al., 2017*). Therefore, to infer bacteria that may drive changes in cancer gene expression we searched for "bottleneck" bacterial nodes that link microbial and inflammatory/antiviral subnetworks using the BiBC metric (*Dong et al., 2015*; *Morgun et al., 2015*; *Thomas et al., 2016*). The top three microbes with highest BiBC represented OTUs from families *Bacillaceae* and *Halobacteriaceae* and the genus *Prevotella* (Fig. 2B).

Due to limitations of 16S rRNA sequencing (limited resolution and sensitivity if compared to metagenomic data analysis (*Poretsky et al., 2014*)), QIIME was unable to assign these OTUs to specific species. However, the *Prevotella* OTU had more advanced assignment than the other two top OTUs, which were assigned only to family level. An alternative bioinformatics approach was therefore used to identify bacterial species that were represented by the *Prevotella*-related OTU. We aligned representative sequences from all OTUs that were collapsed into the *Prevotella* OTU (OTU_97.1949) to the SILVA 16S rRNA database containing sequences of *Prevotella* genus (Fig. 2C). While matches for seven different species from this genus were found, *P. bivia* had the most hits, indicating that this species was the most probable bacterium connected to cancer gene expression. Review of recent literature with publicly available 16S rRNA data of healthy cervix and vagina showed that *Prevotella* had much higher abundance in our cervical cancer cohort than in five studies of healthy vagina (*Gajer et al., 2012*; *Hong et al., 2016*; *Ling et al., 2010*; *Liu et al., 2013*; *Ravel et al., 2011*) and four studies of healthy cervix (*Audirac-Chalifour et al., 2016*; *Di Paola et al., 2017*; *Lee et al., 2013*; *Si et al., 2017*) (Fig. 2D and Table S4). Altogether these results provided *P. bivia* as a candidate bacterium that may contribute to upregulation of proinflammatory/antiviral genes in cervical cancer.

Cervical carcinomas consist of a mixture of stromal, immune, and cancer cells (*Pilch et al., 2001*; *Sheu et al., 2001*). Therefore, the bacteria harbored by tumor might have direct and indirect effects on cancer cells. To test if bacteria predicted by our analysis
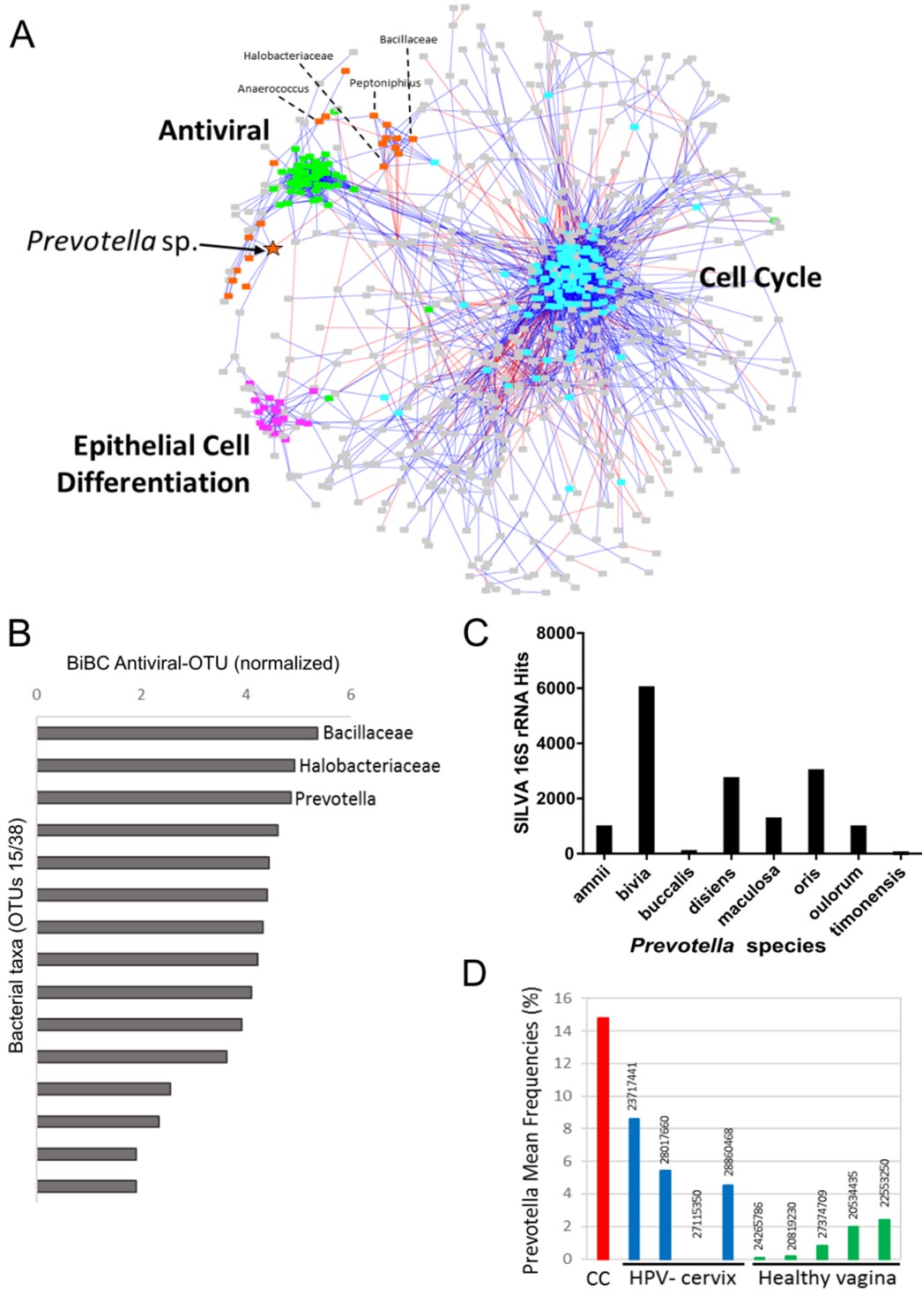

**Figure 2** **Transkingdom microbe-gene regulatory network.** (A) Transkingdom correlation network ($p < 0.001$; FDR < 0.1) between microbial network (21 OTUs, 50 edges) and previously described (*Mine et al., 2013*) tumor differentially expressed genes (698 DEGs, 3,066 edges) connected by 19 edges. Edge-weighted spring layout was performed in Cytoscape. Nodes represent: orange—bacteria; green—antiviral genes; purple—epithelial cell differentiation genes; blue—cell cycle genes; and gray—genes not assigned to specific subnetwork or function. Lines indicate: blue—positive of correlation between nodes; red—negative of correlation between nodes. Orange star indicates *Prevotella* OTU with high BiBC whereas dashed lines connecting the node and its name designate the top five BiBC scored bacteria OTU. (B) Top BiBC OTUs (15/38) calculated between microbial subnetwork and antiviral subnetwork. (C) Top *Prevotella* species in SILVA 16S rRNA database matched to representative sequences assigned to OTU_97.1949 (match length >200 bp, mismatch = 40 bp). (D) *Prevotella* mean abundance in cervical cancer (CC) compared with previous 16S studies for healthy adjacent sites: HPV negative cervix (HPV-cervix) and healthy vaginal microbiome (healthy vagina). (PMID numbers of the source article specified for each column).

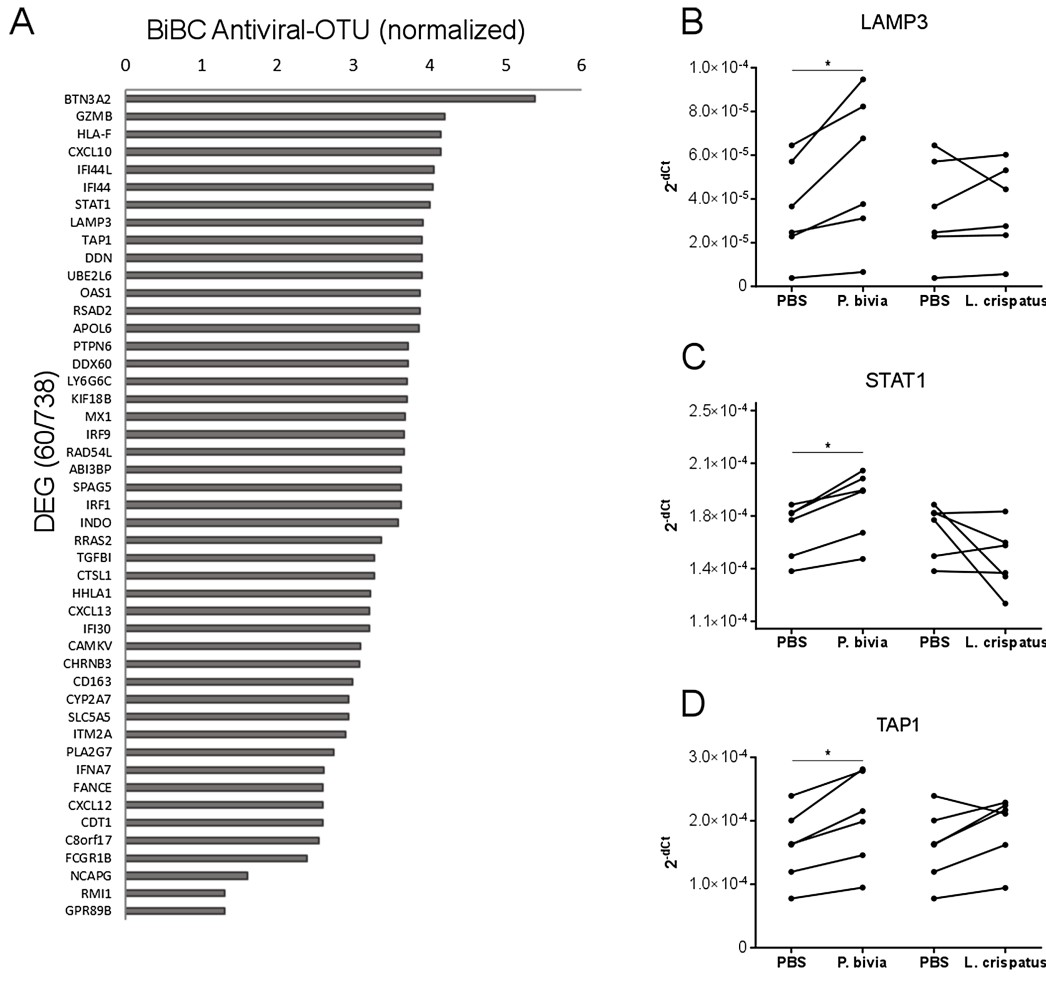

**Figure 3** **Host gene expression regulated by *P. bivia*.** (A) Top DEGs (60/738) ranked by BiBC centrality calculated between bacteria and antiviral genes in transkingdom network (normalized BiBC = log2 ((BiBC*$10^6$) + 1)). (B–D) RT-qPCR for cervical cancer top BiBC genes (LAMP3, STAT1, and TAP1), for which gene expression was upregulated in HeLa cells by *P. bivia* but not *L. crispatus* co-culture compared to negative treatment (PBS). mRNA levels were normalized to 18S rRNA gene expression. (*$p$-value < 0.05, one-tailed Wilcoxon matched-pairs signed rank test).

can drive cervical cancer gene expression program directly, we performed in vitro co-incubation of *P. bivia* with a cervical cancer cell line (HeLa) and measured expression of selected genes. We used *L. crispatus* as a control because it is a common standard in co-culture studies for bacterial induction of innate immunity (*Eade et al., 2012*; *Libby et al., 2008*). While *L. crispatus* was among top most abundant bacteria detected in cancer samples, it showed decreased abundance in cervical cancer compared to other body sights (Fig. S2 and Table S4) and it was not connected to any cancer genes in the transkingdom network (BiBC = 0) indicating no influence on cancer gene expression program. To evaluate which genes might be the most affected by bacteria we calculated BiBC between the microbial portion of the transkingdom network and inflammatory/antiviral subnetwork and selected top 10 genes for further analysis (Fig. 3A). Our gene expression data was generated from whole tissue samples; therefore, it was not surprising that two (GZMB, CXCL10) out of 10 selected genes were not detected in the co-incubation experiments with the cancer cell line. Indeed, GZMB is mostly expressed by T-lymphocytes and natural killer cells (*Bratke et al., 2005*; *Johnson et al., 2003*) and probable sources of CXCL10 expression are endothelial and stromal cells (*Panzer et al., 2006*; *Proost et al., 2003*). Among the eight remaining genes we found that three (LAMP3, STAT1, and TAP1) were upregulated by *P. bivia* (Figs. 3B–3D) whereas no genes were downregulated by this bacterium (Figs. 3B–3D and Fig. S3). These results further support that intra-tumor bacteria like *Prevotella* may be involved in control of the gene expression program of cervical cancer.

## DISCUSSION

The use of transkingdom network analysis in our study provided novel insight into the host-microbiome relationship in cervical cancer. The method has previously proved to be successful for identification of members of the microbial community responsible for a variety of pathological situations, such as enteropathy associated with common variable immunodeficiency (*Shulzhenko et al., 2017*), diabetes (*Greer et al., 2016*), as well as microbes and microbial genes that regulate the effect of antibiotics on the intestine (*Morgun et al., 2015*). Our work demonstrates a new approach where transkingdom network is used to elucidate the pathogenic role of microbiota in cancer. We combined a bacterial co-abundance network with gene expression of key pathways in cervical carcinogenesis (*Mine et al., 2013*), increasing the probability of detecting host-microbiome relationships of relevance for cancer development. The results strongly support a role of microbes from the *Bacillaceae* and *Halobacteriaceae* families and the genus *Prevotella* in regulation of a pro-inflammatory pathway that is activated in cervical cancer (*Mine et al., 2013*). This opens the possibility that bacteria are involved in the elimination of episomal HPV in cervical cancer and thereby play a role in carcinogenesis.

Results from the transkingdom network analysis were further tested and showed that the most promising candidate, *P. bivia,* upregulated three genes (LAMP3, STAT1, and TAP1) in the pro-inflammatory pathway (*Mine et al., 2013*). Noteworthy, LAMP3 is one of the key drivers of this pathway that controls expression of STAT1 and several other antiviral genes (*Mine et al., 2013*). Overexpression of LAMP3 has been shown to promote

metastasis in cervical cancer xenografts and to associate with poor treatment outcome in clinical studies (*Kanao et al., 2005*). Moreover, LAMP3 is induced by hypoxic conditions (*Mujcic et al., 2009*; *Nagelkerke et al., 2013*) and potentially stimulates hypoxia dependent cancer metastasis (*Lu & Kang, 2010*; *Nagelkerke et al., 2013*). Co-culture of a cervical cancer cell line with *P. bivia* in our work showed that this bacterium indeed upregulates LAMP3 under anaerobic conditions. It is therefore reasonable to hypothesize that *P. bivia* (and possibly anaerobes) attracted by hypoxic environment, infiltrates the tumor and induces a pro-inflammatory gene expression program via up-regulation of LAMP3. LAMP3 itself seems to play a crucial role in cervical cancer. Similarly to its role in other cancer types (*Nagelkerke et al., 2011*; *Sun et al., 2014*), it promotes metastases and, by driving expression of multiple antiviral genes, it is potentially involved in elimination of episomal HPV which leads to overexpression of the E6 and E7 HPV oncogenes and disease progression (*Munger et al., 1989*). To better understand the mechanism of LAMP3 mediated effects in cervical cancer further studies are needed that would use techniques of knockdown/knockout and overexpression of LAMP3 in combination with *P. bivia* co-culture.

Our study is a first step toward connecting intra-tumor microbiome to molecular pathways operating in cervical cancer. We analyzed bulk gene expression of the tumor containing a mixture of transcriptomes of different cell types. Therefore, it might not be surprising that genes such as GZMB and CXCL10, which are expressed by non-cancerous cells, were among top ones predicted to be induced by intra-tumor microbiota. Therefore, future studies may take advantage of single cell sequencing allowing for identification of indirect communication between cancer cells and bacteria, for example via infiltrating immune cells. In addition, while our results pointed to several bacteria that might be important, we were able to focus only on one of them, *P. bivia*. It is commonly believed that microbiota may affect its host through combined effect of several taxa rather than as single bacterium (*Lamont & Hajishengallis, 2015*; *Round & Mazmanian, 2009*). Better species identification can be achieved by employing shotgun together with 16S rRNA sequencing since this approach was shown to improve detection and overall diversity of bacterial species (*Dong et al., 2017*; *Ranjan et al., 2016*; *Shulzhenko et al., 2017*). Therefore, whole genome shotgun sequencing should be considered in future studies.

## CONCLUSIONS

A possible involvement of bacteria in the carcinogenesis of cervical cancer could have implications for women presenting with persistent HPV infection. Unraveling the crosstalk between cervico-vaginal microbiota and the woman body may allow development of personalized preventative measures against the infection through shaping the microbiome to favor fast virus elimination (*Kassam et al., 2012*). It could also help to identify single bacterial or community markers of disease progression for better diagnostics. When used together with HPV testing, such markers may lead to more precise evaluation of a woman's risk of developing cancer that would be especially useful in undeveloped countries where access to the healthcare system is limited

(*Denny, Quinn & Sankaranarayanan, 2006*). Additionally, new evidence points out to cervical microbiome as an important player in development (*Zhang et al., 2018a*) and treatment of cervical intraepithelial neoplasia (*Zhang et al., 2018b*) that needs to be investigated and considered. Pinpointing the role of bacteria in treatment outcomes at later stages of the disease (*Iida et al., 2013*; *Muls et al., 2017*; *Vetizou et al., 2015*) could also empower identification of bacterial and host gene targets for new anticancer therapies. Our results open a new direction for the development of companion diagnostics that would evaluate intra-tumor microbiome and guide the administration of antibiotics as an adjuvant to standard therapy.

## ACKNOWLEDGEMENTS

We thank the Center for Genome Research and Biocomputing personnel for support in sequencing and computation infrastructure.

### Funding

This research was supported by startup funds for Andrey Morgun and Natalia Shulzhenko from Oregon State University (OSU), United States, the Norwegian Cancer Society grant nr 107438—PR-2007-0179 (Heidi Lyng), seed grant from College of Pharmacy OSU (Andrey Morgun), DK103761 (Natalia Shulzhenko). The funders had no role in study design, data collection and analysis, decision to publish, or preparation of the manuscript.

### Grant Disclosures

The following grant information was disclosed by the authors:
Oregon State University (OSU).
The Norwegian Cancer Society: grant nr 107438—PR-2007-0179.
College of Pharmacy OSU: DK103761.

### Competing Interests

The authors declare that they have no competing interests.

### Author Contributions

- Khiem Chi Lam conceived and designed the experiments, performed the experiments, analyzed the data, prepared figures and/or tables, authored or reviewed drafts of the paper, approved the final draft.
- Dariia Vyshenska conceived and designed the experiments, performed the experiments, analyzed the data, prepared figures and/or tables, authored or reviewed drafts of the paper, approved the final draft.
- Jialu Hu analyzed the data, authored or reviewed drafts of the paper, approved the final draft.
- Richard Rosario Rodrigues analyzed the data, prepared figures and/or tables, authored or reviewed drafts of the paper, approved the final draft.

- Anja Nilsen analyzed the data, prepared figures and/or tables, authored or reviewed drafts of the paper, approved the final draft.
- Ryszard A. Zielke contributed reagents/materials/analysis tools, authored or reviewed drafts of the paper, approved the final draft.
- Nicholas Samuel Brown performed the experiments, authored or reviewed drafts of the paper, approved the final draft.
- Eva-Katrine Aarnes performed the experiments, authored or reviewed drafts of the paper, approved the final draft.
- Aleksandra E. Sikora contributed reagents/materials/analysis tools, authored or reviewed drafts of the paper, approved the final draft.
- Natalia Shulzhenko conceived and designed the experiments, contributed reagents/materials/analysis tools, authored or reviewed drafts of the paper, approved the final draft.
- Heidi Lyng conceived and designed the experiments, contributed reagents/materials/analysis tools, authored or reviewed drafts of the paper, approved the final draft.
- Andrey Morgun conceived and designed the experiments, contributed reagents/materials/analysis tools, authored or reviewed drafts of the paper, approved the final draft.

### Human Ethics

The following information was supplied relating to ethical approvals (i.e., approving body and any reference numbers):

The clinical protocol was approved by the Regional Committee for Medical Research Ethics in southern Norway (REC no. S-01129). Written informed consent was obtained from all patients.

### DNA Deposition

The following information was supplied regarding the deposition of DNA sequences:

The dataset generated and analyzed during the current study is available in Sequence Read Archive under accession no. SRP131188.

### Data Availability

The OTU tables are provided as a part of the supplementary material (see Table S2). The type of the software and the parameters that were used to generate this data are described in the Materials and Methods section.

### Supplemental Information

Supplemental information for this article can be found online at http://dx.doi.org/10.7717/peerj.5590#supplemental-information.

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
