# Peer review of "Transkingdom network reveals bacterial players associated with cervical cancer gene expression program"

_PeerJ, doi:10.7717/peerj.5590_

## Round 0.1 · original submission · Minor Revisions

Both reviewers found your work interesting and well-written. However, they also found several minor issues. Therefore, please address all critical points raised by both reviewers and revise manuscript accordingly.

Reviewer 1 ·

Basic reporting

Good, especially with a good supporting online material collection.

Experimental design

Good, especially with the in vitro studies supporting the bioinformatics work.

Validity of the findings

This reviewer does not have the expertise to verify all of the bioinformatics work. This reviewer can only attest that apparent standards and practices for due diligence have been applied in this study.

Additional comments

The strengths of this report are basic approach of combining the bioinformatics work with the HeLa cell work, along with the results of same; and the very strong supporting online material collection.

The weakness of this report is that the topic is weell covered in the literature and it is not clear what significant, new unpublished findings are reported here.

As a minor point, “Shannon diversity index” is apparently, usually spelled with two n’s in “Shannon”, not the case for the Y-axis label of supplemental figure 1.

Reviewer 2 ·

Basic reporting

Authors work on transkingdom network reveals bacterial players associated with cervical cancer gene expression program is very well written and they have used professional language with only minor typos/spacing issues

The manuscript literature work is up to date and the background information on cervical cancer, HPV, and general information on bacteria is very well written.

The structure and flow of the manuscript is admirable and logical. Figures, tables and legends were well done with clear labelling and markings.

Authors need to use peerj literature citation standards. Please see the attachment for more comments.

Experimental design

Experimental design to investigate microbiota in cervical cancer is appropriate and also logical to back up their hypothesis.

Authors work on bio-informatics is excellent and they have used all the possible methods that help in screening the key bacteria that might involve in carcinogenesis.

No comments on experimental design other wise.

Validity of the findings

The results obtained by 16 sRNA sequencing is very informative and results are clear. Their work on narrowing down to particular bacterial species and further work on co-culture and identification of key genes is excellent.

These findings are novel and very impactful in future research. Specifically, work on P.bivia which might play an important role in cancer.

Data is robust and the statistical significance is important in these kind of work and authors have done an careful interpretation of results and also pointed out the pit falls of using screening methods.

Authors are encouraged to speculate more on how to use this work on subjects (patients) that are already in treatment and identifying the changes in microbial species in patient samples who are undergoing treatment.

Additional comments

Transkingdom network reveals bacterial players associated with cervical cancer gene expression program

Basic Reporting:

Authors have investigated microbiome involved in cervical cancer by analyzing various bacterial species using bio-informatic and biological approches.

Authors manuscript is clear and written in professional language making it understandable yet scientific. Authors usage of sentence is very constructive and flow of the manuscript is smooth.
However, there are few grammatical mistakes in the manuscript which I have mentioned in the my comments section below. Cervical Cancer is major threat and authors have done an excellent work on elucidating the role of bacteria in cancer carcinogenesis. This kind of work is of huge importance and it helps to identify the bottle neck problems caused by cervical cancer.

Introduction and background:

Here are some major and minor comments:

Major comments:

Authors have done an excellent job in giving the proper introduction regarding cervical cancer, its prevalence and its causes. Authors information on High-risk human papillomavirus (hrHPV) and its key genes that play an crucial role in infection is very impressive and shows that authors are well versed in this particular field.

Background information on the cervical cancer, viral infection, key genes, and problems related to diagnosis and cure were accurately addressed. The flow of introduction is well written where in the authors mentioned about cancer and how they wanted to address some key aspects by microbial linkage is impressive. The role of microbiota is crucial and it is important to learn and study about the particular microbes that play a telling role in cancer diseases.

Literature work:

Authors have done an excellent job in citing the appropriate literature work. However, there need to be a substantial correction in the way the citation needed to be written as per the standards of this journal. Please follow the guidelines from the link below.

https://peerj.com/about/author-instructions/#standard-sections

Authors need to remove parenthesis from the citation and required to give the year of the publication. This has to be changed in the entire manuscript to match the journal style of citations.


Minor comments:
Authors are encouraged to rephrase the sentence starting from line 82-87. The representation of cited work for each case/health related areas next to each other. It is understandable that they wanted to cite work according to the area however this inclusion at each step is creating a confusion when trying to read the whole sentence. It will logical to mention about the microbiota role in several diseases/areas of health and provide all the citations at the end in order from 1st to last. This way readers can understand the sentence faster and if need references can look for them at the end.

Authors are encouraged to remove the word “like” in line 91 and change it to following sentence.

Reviewer suggestion:

“Studies of vaginal (Champer et al.) and cervical (Vyshenska et al.) microbiota
have found association between changes in microbial community, overall diversity, abundances of particular taxa, and cancer development.”
Authors are encouraged to replace the word “power” from line 100 to “significance”. The word significance will also boost the authors innovative approach to tackle the fatal cancer and yes indeed, this approach is of great significance in future research related to many other diseases.



Materials & Methods

Major comments:

Authors approach to analyze the microbiota in relation to cervical carcinogenesis is standard and appropriate. Authors have used several bio-informatic approaches to assign, correlate, diversify different species of bacteria. No of patients specimens (123) used for the analysis is adequate for this study. Authors methods such as 16s rRNA sequencing, gene expression, cell culture techniques are relevant and appropriate for this particular study. Authors used proper controls for in-vitro studies to investigate cancer gene expression in relation to P.bivia.

Minor comments:

Please check for spacing in Line 148.

Results:

Major comments:

Overall results from bio-informatic data especially 16 sRNA sequencing results gave very good information on the variation of microbiota in cervical cancer samples compared to that of healthy cervix, vagina, stool, or skin sample. However, It is not an novel finding as many other groups have also found similar results.

Authors work on building transkingdom network by integrating microbial abundance and cancer gene expression is very logical and the results are very informative.. Alternative approaches to identify different species from genus provetella results are successful where authors have found high abundance of P.bivia.

Authors findings of key genes that are upregulated by P. bivia from whole tissue samples is a new finding and it is interesting to perform future experiments relating to protein level and biochemical characterization of these particular genes to understand their clear role in carcinogenesis.

Minor comments:

Authors mentioned in line 298 about limitation of 16sRNA Sequencing in assigning OTUs to specific species. It will be encouraging to give more details about this limitations. This is unclear as there is an high BiBC results of Bacillaceae and Halobacteriaceae is more than Prevotella.

Figures and tables:

Authors have provided excellent figure to confirm their findings. All figures are very well organized and legends are written with adequate descriptions. Tables and supplementary figures also support the important findings. All figures and tables were well organized and clearly labeled.

Discussion:

Major comments:

Discussion was well written with supporting evidences from the results and also mentioning the key points in microbial involvement in cervical carcinogenesis. Discussion part for future work on species identification can be elaborated specifically, how shot gun method can improve the studies in cervical cancer microbiota.

Minor comments:

I thank you for providing the critical information about microbial involvement in cancer. However, it is interesting to see the microbiota data from subjects that are already involved in treatment. Please see the new publication related to microbiota in patients involved in cervical cancer that are involved in treatment. I am hoping this will be helpful for future research and you can also cite this paper if you have any valuable information.

Zhang, Hongwei & Lu, Jiaqi & Lu, Yingying & Cai, Qingqing & Liu, Haiou & Xu, Congjian. (2018). Cervical microbiome is altered in cervical intraepithelial neoplasia after loop electrosurgical excision procedure in china. Scientific Reports. 8. 10.1038/s41598-018-23389-0.

Authors are encouraged to look into one of the latest studies on microbiota effects on cervical intraepithelial neoplasia (CIN) from Zhang et al (2018) group where they have shown microbes that might play an important role in carcinogenesis. Authors are also encouraged to discuss on prospects about characterization of specific proteins that involve in cancer carcinogenesis. Athours have already narrowed down 3 key genes from P.bivia that are upregulated and might play important role in cancer. Can authors suggest any follow up experiments to investigate how exactly LAMP3 functions?

Overall authors have done an excellent work on identifying specific bacteria that may have an effect on cancer and they have conducted their studies with logical and scientific way in taking advantage of bio-informatic tools.

---

## Round 0.2 · accepted · Accept

All critical points were addressed and the manuscript was revised accordingly. Therefore, the amended version can be accepted.

#